# Nutritional Profiles of Non-Dairy Plant-Based Cheese Alternatives

**DOI:** 10.3390/nu14061247

**Published:** 2022-03-16

**Authors:** Winston J. Craig, A. Reed Mangels, Cecilia J. Brothers

**Affiliations:** 1Center for Nutrition, Healthy Lifestyles, and Disease Prevention, School of Public Health, Loma Linda University, Loma Linda, CA 92354, USA; 2Vegetarian Resource Group, Baltimore, MD 21203, USA; reedmangels@gmail.com; 3Department of Biology, Walla Walla University, College Place, WA 99324, USA; cecilia.brothers@wallawalla.edu

**Keywords:** non-dairy cheese alternatives, plant-based cheese alternatives, vegan, calcium, protein, vitamin B12, vitamin D, sodium, saturated fat

## Abstract

With the growing interest in non-dairy products, there has been a surge of interest in consumers seeking plant-based cheese alternatives spurred by a desire to improve individual health and achieve a more sustainable food supply. The aim of this study was to conduct a cross-sectional survey of non-dairy cheese alternatives available in the United States and to evaluate their nutritional content. A total of 245 non-dairy plant-based cheese alternatives were analyzed using their nutritional facts labels. The various cheese alternatives were based upon coconut oil (*n* = 106), cashews and coconut (*n* = 61), cashews (*n* = 35), oats (*n* = 16), almonds (*n* = 7), soy (*n* = 6), palm fruit oil (*n* = 5), and other blends (*n* = 9). Only 3% of these cheese alternatives had 5 g or more of protein, while 19%, 14%, and 1% were fortified with calcium, vitamin B12, and vitamin D, respectively. Almost 60% had high levels of saturated fat, while 15% had low sodium levels. The products based on cashews alone more commonly had the highest protein levels and the lowest sodium and saturated fat levels. Those containing coconut oil more commonly had higher saturated fat and sodium levels and were most frequently fortified with vitamin B12. Few of these products could be considered good dietary sources of either protein or calcium.

## 1. Introduction

For many people, cheese is a favorite food item, in addition to being an important ingredient in favorite foods, such as pizza, lasagna, enchiladas, and cheeseburgers. Traditionally, cheese has been made with milk from a variety of animals, primarily from cows, but also from goats, sheep, water buffalo, and others. As interest in reducing one’s consumption of animal-based products has grown [1], plant-based cheese alternatives based on nuts, oils, grains, soy, and other plant products have been developed. Plant-based cheese alternatives are produced using techniques similar to those used in the production of processed dairy cheese [2]. An emulsion is created which may include oils and protein from plant sources, water, emulsifiers, stabilizers, and natural flavorings [2,3]. Some products may also contain preservatives, such as olive extract or sorbic acid, and coloring agents, such as beta-carotene or annatto [2]. Potato starch provides anti-clumping properties, while tapioca starch adds elasticity to the product. Processes such as applying heat, acids, or enzymes may be used to improve product texture [3]. Powdered cellulose is used as an anti-caking agent. The resulting plant-based cheese alternatives are sold in various forms, including as blocks, slices, shreds, chunks, wedges, and spreads. 

Consumers have many reasons for choosing plant-based cheese alternatives. These can be related to health concerns such as lactose intolerance or a milk allergy, to concerns about the environment and animal welfare, or to a yearning for variety in their diets that inspires them to try new products [4]. The increase in the number of people following vegan diets and with an interest in vegan foods is also likely to be a factor in the increased production of and demand for plant-based cheese alternatives. A survey conducted in 2020 found that 25% of adults in the United States always or sometimes eat vegan meals when eating out and that 3% of adults in the United States follow a vegan diet [5]. 

In the United States, in 2020, plant-based cheese sales totaled $270 M, which was a 42.5% increase over the previous year, making the category of plant-based cheese alternatives one of the fastest growing categories of plant-based foods [6]. In 2021, the market for vegan cheese was forecast to grow 11.8% between 2021 and 2028 [7]. The global market for vegan cheese was valued at 2.22 billion USD in 2020 and was forecast to grow at a compound annual growth rate of 12.4% between 2021 and 2028 [7]. 

Recent research has examined the nutritional content of non-dairy plant-based beverages (also known as plant-based milks) and plant-based yogurt alternatives [3,8,9,10,11,12,13,14]. These products are based on various grains, nuts, seeds, legumes, and fruits and contain a significant variety of protein and nutrient fortification levels. The nutritional composition and attributes of plant-based cheese alternatives have been examined in products purchased in Europe, Spain, and the United Kingdom [3,14,15]. In the United Kingdom, some of these products are fortified with nutrients including calcium, vitamin D, and vitamin B12 [14]. In view of the large market for these products in the United States, an assessment of plant-based cheese alternatives available to United States consumers is needed. 

The primary aim of this study was to evaluate a cross-sectional survey of the nutritional content of plant-based cheese alternatives that are commonly available in the United States. Secondary aims included the evaluation of the effect of product composition on nutritional quality and the determination of the extent and type of fortification. This study only examined products that were entirely plant-based. This research will be useful to healthcare professionals and consumers and will indicate possible areas for additional research on product composition and on the modification of the nutritional quality of plant-based cheeses.

## 2. Materials and Methods

A total of 295 plant-based cheeses were identified by visiting supermarkets and health food stores in the western United States and by searching the websites of the manufacturers of these plant-based cheeses. Dips, sauces, and spreads were not included in our analysis. Products reporting casein as a minor component on their ingredient list were also excluded. Products available for purchase online that had no retail outlet stores in the United States were not included in our analysis. Products that are a hybrid of dairy products and plant products [2,16] were also excluded. Similarly, products with incomplete nutritional data were excluded from the study. The ingredients and nutritional contents of 245 plant-based cheese alternatives, representing 35 brands, were recorded, using either the nutritional facts label on the retail package or the data collected from the website of the manufacturer. The data representing the 245 cheeses were collected from November 2021 to January 2022. The nutrients per serving size included calories, fat, saturated fat, sodium, carbohydrates, dietary fiber, total sugars, protein, calcium, vitamin D, and vitamin B12. The cheeses were divided into groups based upon their major ingredients. The median values of all nutrients were calculated for each type of cheese. The levels of fortification for calcium, vitamin D, and vitamin B12 were calculated for each group of cheese.

The nutritional value of each plant-based cheese was rated according to the following criteria: calcium, vitamin D, and vitamin B12 fortification and at least 5 g of protein/serving (10% of the Daily Value or DV). The health qualities demonstrated by the ingredients were determined by the following criteria: not more than 1 g of saturated fat/serving (5% of the DV); not more than 115 mg sodium/serving (5% of the DV); 230 mg or more of sodium/serving (10% of DV); not more than 100 calories/serving (5% of 2000 calorie intake); at least 1.5 g of dietary fiber/serving; and 4 g or less of fat/serving (5% DV). In addition, we noted how many cheeses had high levels of fat and saturated fat. This was tabulated as the number of cheeses having 15.5 g or more of fat/serving (20% DV for fat) and having 4 g or more of saturated fat/serving (20% DV for saturated fat). The US Dietary Guidelines specify, as a general guide, that 20% DV or more of a nutrient/serving is considered high, while 5% DV or less of a nutrient/serving is considered low [17,18]. In the United States, the DV for sodium is 2300 mg, for fat is 78 g, for protein is 50 g, for saturated fat is 20 g, and for dietary fiber is 28 g [18]. For the purposes of this study, we suggest that the sodium and saturated fat levels in the cheeses should not exceed 5% of their DV (a designated low level), while protein should have a minimally acceptable level of at least 10% of the DV/serving (5 g). Ten percent was chosen as a mid-stream number between the 5% DV (low value) and the 20% DV (high value). 

### Statistical Analysis

Data were tested for normality and homoscedasticity prior to analysis. The median and interquartile ranges were used for descriptive statistics, as the data were not normally distributed. Nutrients were compared across the different bases of non-dairy cheeses using a Kruskal–Wallis test for each nutrient, followed by Dunn’s post hoc test with Bonferroni adjustment for multiple comparisons. A significant *p*-value < 0.05 was used for all analyses. R software was used to conduct all statistical analyses [19].

## 3. Results

The 245 non-dairy plant-based cheese alternatives analyzed were based upon almonds (*n* = 7); cashews (*n* = 35); cashews and coconut oil (*n* = 61); coconut oil with food starch (potato, tapioca, and/or corn starch) (*n* = 106); oats (*n* = 16); soy and coconut oil (*n* = 6); and the following mixtures: palm fruit oil and corn and/or potato starch (*n* = 5), potato starch and canola oil (*n* = 3), tapioca flour and canola/safflower oil (*n* = 2), soy or other vegetable oil (*n* = 3), and coconut and oats (*n* = 1). Table 1 displays the medians of the calories and each nutrient for each of the base types. Any significant differences among the base types are also reported. Several brands had multiple varieties of non-dairy cheese alternatives that had very similar ingredient lists, except for the herbal seasonings and flavorings.

The most common forms of cheeses were blocks and wheels (27%), shreds (24%), and slices (20%). While there were a variety of unique and exotic flavors, the most common were cheddar (18%), mozzarella (14%), pepper jack (5%), and smoked gouda (4.5%). Of the 35 brands examined, only 12 brands (34%) had calcium fortification, 4 (11%) had vitamin B12 fortification, and 1 (3%) had vitamin D fortification. Twenty-two brands (63%) had no fortification at all. Of the 245 varieties, 183 (75%) had 2.5 g or less (5% DV) of protein.

Seven of the twelve brands that had calcium fortification (58%) used tricalcium phosphate, two brands (17%) used calcium citrate, two brands (17%) used calcium sulfate, and one brand (8%) used a tricalcium phosphate–calcium sulfate mix to achieve calcium fortification. The typical fiber added to many of the cheeses was methylcellulose, an anticaking agent. One serving size of cheese varied from 14 to 40 g depending upon the format of the cheese. The most common serving sizes for block cheeses and shreds were 28 and 30 g, while slices typically were 20 g. 

Table 2 summarizes the data showing the percentage of the plant-based cheese alternatives that contain (a) a reasonable level of important nutrients; (b) low levels of sodium, calories, fat, and saturated fat; and (c) high levels of sodium, fat, and saturated fat. Additionally, in Table 2, the data for the cheese alternatives that meet or exceed suggested nutrient guidelines are separated according to the different bases for comparison.

The top seven companies that had the highest sales from non-dairy plant-based cheeses in 2020 represented 41% of the market share of cheese products in our study. Yet, despite their leadership in sales, none of their products had either vitamin D fortification or at least 5 g of protein per serving, only 1 in 3 had any calcium fortification, and only 1 in 4 had vitamin B12 fortification (the majority of which originated from the same brand). 

## 4. Discussion

Regular dairy cheese is typically considered a good source of protein and calcium. When people choose a plant-based non-dairy cheese alternative, they often expect a product that has a similar taste, texture, appearance, and nutrient profile and similar functional characteristics (such as melt performance) to a regular dairy cheese product [3,6,20]. However, the median values for the protein and calcium content of the non-dairy cheeses in our study were found to be zero (Table 1). Only 3% of the products reached a level of 5 g of protein per serving, while less than 20% of the cheese alternatives were fortified with calcium (Table 2). Of the top seven selling brands of non-dairy cheese alternatives, representing 41% of all varieties of cheese alternatives, only one-third of the products had calcium fortification (Appendix A) [20]. Considering all 245 cheeses, less than 1 in 5 had calcium fortification, less than 1 in 7 had vitamin B12 fortification, and only 1% had vitamin D added (Table 2). In an analysis of 109 plant-based cheese alternatives available in the UK, the authors observed similarly low levels of fortification [14]. Only 13% of the products had calcium fortification and only 37% had vitamin B12 fortification, while barely 2% were fortified with vitamin D. The reported values for the protein content of the plant-based cheese alternatives in the UK were also very low, with less than 0.5 g/30 g serving for coconut oil-based products and less than 2 g protein/30 g serving for nut- and seed-based products [14]. 

A survey of dairy cheeses, in a variety of formats [21], found that dairy cheese typically contains approximately 5–8 g protein per serving and 10–20% DV of calcium, which is substantially more than the non-dairy alternatives (Appendix A). While many dairy beverages and yogurts, along with some non-dairy beverages and non-dairy yogurt alternatives, are fortified with vitamin D [9], dairy cheeses typically are not fortified with vitamin D [22]. 

In a Spanish study of 40 plant-based cheese alternatives found in local supermarkets, the coconut oil-based products (accounting for 85% of the products) had close to zero protein, while the cashew and tofu-based cheese alternatives reported median levels of approximately 3 g and 5 g of protein/30 g serving [15]. In their study, fortification of the cheese products was not addressed. In our study, the cashew-based cheese alternatives were more likely to have higher protein levels than the coconut-based products. While a number of the cheese alternatives in our study contained chickpea-miso or faba-bean protein, the amounts added were insufficient to significantly impact the protein content. Furthermore, the coconut oil-based cheeses were more likely to be fortified with vitamin B12 (Table 1 and Table 2). 

We found that only one in seven of the non-dairy cheese alternatives had low saturated fat levels and one in seven had low sodium levels (Table 2). In addition, 98% of the products had only 1 g of sugar or less. The almond and cashew-based products were the lowest in saturated fat content (0–1 g/serving), while those containing coconut were rich in saturated fat (4–7 g/serving) (Table 1 and Table 2). The cashew-based cheese alternatives were the most likely products to be low in sodium (Table 1). Over 80% of the cheese alternatives were low in calories, while only 2% were high in fat. Almost 60% of the products had high saturated fat levels (due to the rich presence of coconut oil), while almost one in three had elevated sodium levels. The prevalence of coconut oil in the cheese alternatives is of concern given the research data associating its use with the elevation of blood lipid levels [23,24]. 

Excepting the fortification issues, cashew-based cheeses appeared to have a better nutritional profile. Almond paste and peanut oil are apparently emerging as important players in the future of the cheese-alternative market [3]. They can be used to produce a cheese alternative of comparable texture to one made from coconut oil, but with a lower saturated fat content. 

In a survey of United States cheeses [21], the authors report that dairy and non-dairy cheese alternatives are equally likely to show elevated levels of saturated fat, while non-dairy products are twice as likely to have high sodium levels (Appendix A). Another research group analyzed the 114 new vegan cheese alternatives that were launched into the European market in 2020. Compared to the 115 new dairy cheeses, the vegan cheese alternatives were lower in salt and protein but higher in fat and saturated fat due to their coconut oil and palm oil content [3]. Only 10.5% of the vegan cheese alternatives claimed to be free of palm oil, while 26% of the products claimed to be fortified, and only 10.5% had added calcium [3]. 

Interest has been expressed in developing healthier plant cheeses based on soy or other legumes, possibly in combination with grains such as oats [2,25,26]. Soy offers several benefits including its nutrient profile [27], although concerns have been expressed about its beany flavor [26,28]. Additionally, its grainy texture [26] and difficulty in producing hard-type cheeses due to the unsatisfactory coagulation of soy milk have been reported [2]. Other legumes have similar issues [25]. 

A variety of techniques have been proposed to overcome these issues with soy and other legumes. These include the development of a bean that lacks lipoxygenases which have a negative effect on the flavor and odor of products made from soy and other legumes; adding spices and other natural flavoring agents; and using processing techniques to reduce phytates and other undesirable factors [25,26,29]. Fermentation may also be a way to improve the texture and flavor of plant cheeses [2,28,29,30]. The use of legume protein isolates or protein concentrates has also been suggested as a way to improve the color, flavor, and nutrient availability of plant cheeses made from legumes [25]. Combining soy milk with other plant milks such as peanut milk may improve the overall acceptance of plant-based cheese alternatives composed of these combinations [29]. 

Another approach could be to accept the flavor and properties of soy and legume-based cheese alternatives and to recognize that these products are different from dairy cheese [25,29]. This approach may not succeed with consumers who are ultimately looking for a product that mimics dairy cheese, but it may appeal to consumers who appreciate the flavor of legumes or other plant foods in their own right and who appreciate the potentially less processed nature of these products.

We recognize that our research has some limitations. Manufacturers frequently alter the ingredients and thereby the nutrient content of their products, such that a repeated investigation may produce slightly different results. We relied on the information available on product labels and manufacturer websites and did not analyze the nutritional content of the products surveyed. One of the strengths of our research is that it is a cross-sectional survey of the nutritional content of plant-based cheese alternatives that are commonly available in the United States.

## 5. Conclusions

While the demand for non-dairy cheese alternatives continues to rise in the United States, there appears to be a need for products with a better nutritional content and a healthier profile. The marketplace is dominated by non-dairy products containing coconut oil, which results in many products containing very little protein and many having a high saturated fat content. European studies conducted to date reflect the same picture [14,15]. Few cheese-alternative products are fortified with the important nutrients that are provided by dairy cheese—namely, calcium and vitamin B12. Both calcium and vitamin B12 are critical nutrients for vegans, since both nutrients may be marginal in vegan diets [31,32,33]. More widespread fortification with calcium, vitamin D, and B12 would bring a serving of non-dairy cheese alternatives more in line nutritionally with the better fortified plant-based beverages and yogurts [8,9,10]. There is a need for more plant-based cheese alternative options made from legumes, or that have been fortified with a legume. This is especially important where these non-dairy cheese alternatives are fed to young children who require quality nutritional products for proper growth. The near-zero level of protein in most non-dairy plant-based cheese alternatives is particularly worrisome when these products are fed to children with the assumption that they are a good nutritional substitute for dairy cheese. At the present time, non-dairy cheese alternatives should not be considered as a nutritional replacement for dairy cheese. In our experience, the products based on cashews alone were found more commonly to have the highest protein levels and lowest sodium and saturated fat levels.

## Figures and Tables

**Table 1 nutrients-14-01247-t001:** The median (Q1–Q3) values of calories and 10 nutrients of non-dairy cheese alternatives/serving classified according to their bases.

Nutrient	All	Almond	Cashew	Cashew and Coconut	Coconut	Oat	Soy and Coconut	Others ^1^	*p*-Value
*n*	245	7	35	61	106	16	6	14	
Calories	80 (60–100)	70 (60–70) ^abc^	90 (65–120) ^ab^	100 (80–120) ^a^	70 (60–80) ^c^	70 (70–83) ^bc^	110 (110–110) ^abc^	80 (80–90) ^abc^	*p* < 0.001
Fat	7 (5–8)	6 (6–6) ^abc^	7 (5–10) ^abc^	8 (7–11) ^a^	6 (5–7) ^bc^	5 (5–6) ^b^	10 (10–10) ^ac^	6 (6–7) ^abc^	*p* < 0.001
Saturated Fat	4 (2–5)	0 (0–0) ^ab^	1 (1–2) ^a^	4 (3–5) ^cd^	5 (4–6) ^c^	4 (1–5) ^bcd^	8 (8–8) ^c^	3 (2–4) ^abd^	*p* < 0.001
Sodium	190 (150–240)	180 (125–190) ^ab^	130 (98–190) ^a^	150 (110–200) ^a^	215 (180–258) ^b^	200 (200–270) ^b^	205 (190–220) ^ab^	270 (193–298) ^b^	*p* < 0.001
Carbohydrates	5 (4–6)	3 (3–4) ^a^	5 (3–7) ^ab^	5 (4–6) ^ab^	5 (4–6) ^b^	5 (4–5) ^ab^	3 (3–3) ^a^	7 (5–7) ^b^	*p* < 0.001
Fiber	0 (0–1)	1 (1–1) ^a^	1 (0–1) ^ab^	0 (0–1) ^abc^	0 (0–0)^d^	0 (0–0) ^cd^	0 (0–0) ^cd^	0 (0–1) ^bcd^	*p* < 0.001
Sugars	0 (0–0)	1 (1–1) ^a^	0 (0–1) ^b^	0 (0–0) ^b^	0 (0–0) ^c^	0 (0–0) ^bc^	0 (0–0) ^bc^	0 (0–0) ^bc^	*p* < 0.001
Protein	0 (0–3)	2 (2–2) ^abc^	3 (2–4) ^a^	3 (1–3) ^ab^	0 (0–0) ^d^	0.4 (0–3) ^bc^	1 (1–2) ^abcd^	0 (0–1) ^cd^	*p* < 0.001
Calcium	0 (0–0)	0 (0–0) ^abc^	0 (0–0) ^a^	0 (0–0) ^ab^	0 (0–5) ^ab^	0 (0–10) ^abc^	0 (0–0) ^bc^	20 (0–20) ^c^	*p* < 0.001
Vit. D	0 (0–0)	0 (0–0)	0 (0–0)	0 (0–0)	0 (0–0)	0 (0–0)	0 (0–0)	0 (0–0)	0.681
Vit. B12	0 (0–0)	0 (0–0) ^ab^	0 (0–0) ^a^	0 (0–0) ^a^	0 (0–19) ^b^	0 (0–0) ^ab^	0 (0–0) ^ab^	0 (0–0) ^ab^	*p* < 0.001

Different lowercase letters in the same row indicate significant differences between base types. “All” was not included in analyses. *p* < 0.05 is considered statistically significant. ^1^ Palm fruit oil and corn and/or potato starch (*n* = 5), potato starch and canola oil (*n* = 3), tapioca flour and canola/safflower oil (*n* = 2), soy or other vegetable oil (*n* = 3), coconut and oats (*n* = 1).

**Table 2 nutrients-14-01247-t002:** Percentage of non-dairy cheese alternatives meeting or exceeding the suggested guidelines per serving, listed according to the base type and for the total number of cheese alternatives.

	Total	Almond	Cashew	Cashew and Coconut	Coconut	Oat	Soy and Coconut	Others ^1^
*n*	245	7	35	61	106	16	6	14
At least								
At least 5 g protein (10% DV)	3	0	23	0	0	0	0	0
Calcium fortification	19	0	0	7	26	38	17	57
Vitamin D fortification	1	0	0	0	3	0	0	0
Vitamin B12 fortification	14	0	0	5	28	0	0	0
No more than								
115 mg sodium (5% DV)	15	29	34	31	2	0	0	7
1 g saturated fat (5% DV)	14	100	54	0	0	38	0	21
4 g of fat (5% DV)	4.5	0	6	0	7	6	0	7
100 calories (5% DV)	81	86	66	67	93	100	17	7
High levels								
4 g or more saturated fat (20% DV)	57	0	0	59	87	44	83	7
More than 15.5 g fat (20% DV)	2	0	0	10	0	0	0	0
230 mg sodium or more (10% DV)	32	0	9	13	46	44	17	71

^1^ Palm fruit oil and corn and/or potato starch (*n* = 5), potato starch and canola oil (*n* = 3), tapioca flour and canola/safflower oil (*n* = 2), soy or other vegetable oil (*n* = 3), coconut and oats (*n* = 1).

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
