# Peer review of "Nutritional Profiles of Non-Dairy Plant-Based Cheese Alternatives"

_nutrients, 2022, doi:10.3390/nu14061247_

Round 1

Reviewer 1 Report

Thanks to submitted "Nutritional Profiles of Non-Dairy Plant-Based Cheese Alternatives" to Nutrients.  Not general, or study is interesting and important for the readers of Nutrients.  A great concern is that these chosen brands really represent the commercial brands available in all the country.

 Line#67: change "is" to "was"

 Lines#77-80: please rephrase to improve clarify

 Line#121: sunflower

 Line#140: seven of the 245 corresponds to 58%.  Something doesn't fit.

 Line#159-160: How were the top 7 companies established?  Was any reference used?

 Line#159-160: How are the top 7 companies established?  Was any reference used?

 Line#167: I certify that the people seek a similar product also with technological characteristics

 Line#204: healtheir based on?

 Line#220: what vegetables?  Some reported internet this study?

Author Response

Thank you for your thoughtful review. Our responses are attached in the pdf file

Reviewer 2 Report

The manuscript aims to evaluate the nutritional profile of plant-based cheese products sold on the US market.

The manuscript is of interest for the topic, despite I think that the main drawback here is the missing of comparison of the nutritional profiles with the animal milk one. So, this would “teach” people that such drinks are not alternatives to milk, for their composition.

Line 29: I do believe cheese is an important food that can be considered to be eaten as it is, not only as ingredient.

Line 33: “plant cheese”. I guess it should be uniformed the name of such products along the manuscript. More, I do believe the definition given in the title is not correct, as “non-dairy” and “plant-based” are redundant together. I suggest just “plant-based cheese”.

Lines 90-100: how this criteria were set up? is there any reference to some regulation on mandatory/voluntary information where authors picked up information?

Lines 165-167: this information should be supported by a reference and, concerning to the quantity, it should also be filled within a food group (es. animal or vegetal foods?).

I suggest to merge tables 2 and 3 in a unique table.

Tab 1: a) calories is not the correct term, as it is the measure unit. So, it is “energy”, and it should be expressed as kJ and kcal. b) please check significant digits. c) did authors found on the pack salt or sodium?

Author Response

Thank you for your thoughtful review. Our responses to your detailed review are attached in the pdf file. Thank you.